# DSGRec: dual-path selection graph for multimodal recommendation

Zihao Liu[1,*] and Wen Qu[2,*]

[1] College of Computer Science and Technology, Dalian Martime University, Dalian, Liao Ning, China
[2] College of Computer Science and Artificial Intelligent, Liaoning Normal University, Dalian, Liao Ning, China
* These authors contributed equally to this work.



## ABSTRACT

With the advancement of digital streaming technology, multi-modal recommendation systems have gained significant attention. Current graph-based multi-modal recommendation approaches typically model user interests using either user interaction signals or multi-modal item information derived from heterogeneous graphs. Although methods based on graph convolutional networks (GCNs) have achieved notable success, they still face two key limitations: (1) the narrow interpretation of interaction information, leading to incomplete modeling of user behavior, and (2) a lack of fine-grained collaboration between user behavior and multi-modal information. To address these issues, we propose a novel method by decomposing interaction information into two distinct signal pathways, referred to as a dual-path selection architecture, named Dual-path Selective Graph Recommender (DSGRec). DSGRec is designed to deliver more accurate and personalized recommendations by facilitating the positive collaboration of interactive data and multi-modal information. To further enhance the represetation of these signals, we introduce two key components: (1) behavior-aware multimodal signal augmentation, which extract rich multimodal semantic information; and (b) hypergraph-guided cooperative signal enhancement, which captures hybrid global information. Our model learns dual-path selection signals *via* a primary module and introduces two auxiliary modules to adjust these signals. We introduce independent contrastive learning tasks for the auxiliary signals, enabling DSGRec to explore the mechanisms behind feature embeddings from different perspectives. This approach ensures that each auxiliary module aligns with the user-item interaction view independently, calibrating its contribution based on historical interactions. Extensive experiments conducted on three benchmark datasets demonstrate the superiority of DSGRec over several state-of-the-art recommendation baselines, highlighting the effectiveness of our method.

# INTRODUCTION

The exponential growth of multimedia data on network media platforms has underscored the importance of multi-modal recommender systems (MRS) in filtering and delivering relevant information from vast datasets, garnering significant attention from both

Corresponding author
Wen Qu, quwen@skyarch.cn

academia and industry. Unlike traditional recommendation systems that rely solely on user-item interaction data, MRS leverage multi-modal content information (*e.g.*, visual, textual, and auditory attributes) to provide more accurate and personalized recommendation. Recent advancements in graph convolutional networks (GCNs) (*He et al., 2020*) have further enhanced the performance of MRS by enabling the modeling of high-order relationships between user and item nodes (*Wei et al., 2020*). These developments have not only improved recommendation accuracy but also opened new avenues for exploring complex dependencies within multi-modal data.

In the field of MRS, three primary paradigms have emerged for representing users and items: collaborative filtering (CF)-based embedding, which captures user-item interaction patterns; multi-modal feature (MF)-based embedding, which utilizes content information such as images, text, and audio; and the hybrid of the CF-based and MF based features. Early approaches, such as matrix factorization (*Chen, Fang & Saad, 2009*), project user and item IDs into a shared latent space but often fail to capture high-order relationships. Subsequent studies, including LightGCN (*He et al., 2020*), graph convolutional matrix completion (GC-MC) (*Berg, Kipf & Welling, 2017*), and Neural graph collaborative filtering (NGCF) (*Wang et al., 2019*), have advanced CF by organizing user-item interactions as graphs and aggregating multi-hop neighbor information to enrich representations. On the other hand, methods like multi-modal graph convolution network (MMGCN) (*Wei et al., 2019*) and GRCN (*Wei et al., 2020*) integrate multi-modal features into graph structures to enhance user preference modeling. Recent works, such as MICRO (*Zhang et al., 2022a*) and BM3 (*Zhou et al., 2023b*), further bridge the gap between CF and MF by leveraging multi-modal content to enhance user-item interactions.

Despite these advancements, we argue that existing graph-based methods still fail to fully capture the complexity of user-item interactions or address the limitations of fine granularity fusion with two signals in different context. Specifically, there are two key limitations in the exploration of user behavior:

1. Single-path modeling of interactive information limits the learning of complex interactions. As shown Fig. 1 (left), each user interaction is influenced by both demand and the preference. CF signals matching user demand through discover items that similar users have interacted with, while MF information captures associations between items through the alignment and fusion of different modality contents. For example, in Fig. 1 (right), the second user is recommended a dark blue pleated midi skirt based on collaborative signals but ultimately chooses a dark blue striped knee-length skirt with the same style due to preference signals. This indicates that modeling user-item interaction as a single-path fails to comprehensively capture user behavior.

2. The collaboration of demand signals and preference signals lakes adaptability across different contexts. Existing methods can be broadly categorized into two approaches, as shown in Fig. 2, each with inherent limitations. The first approach (Fig. 2, left) separately learns CF-based ID embeddings and MF-based semantic embeddings, subsequently fusing them for recommendation. However, this method often suffers

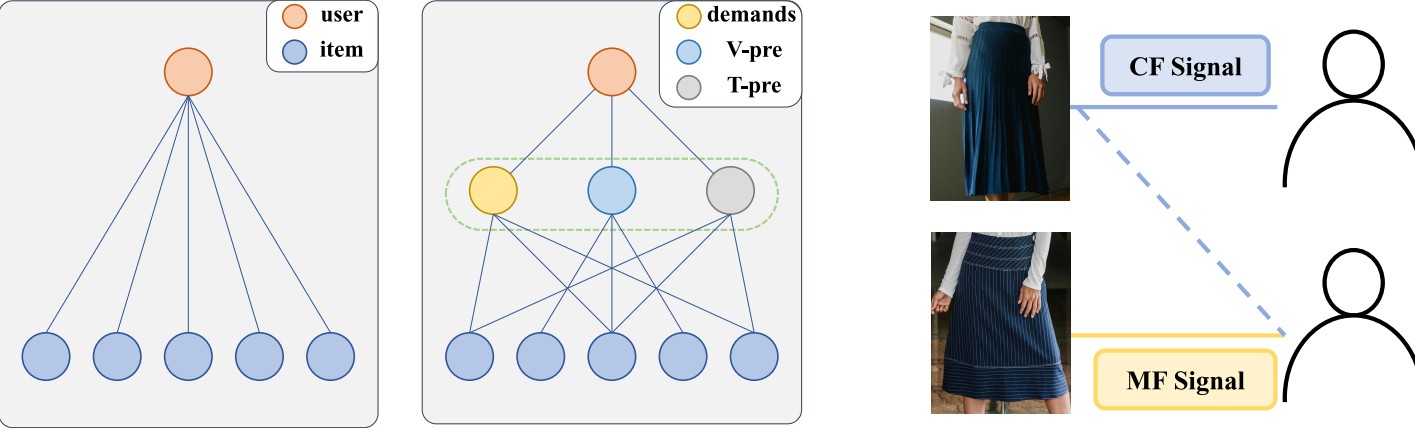

**Figure 1 Specification of diverse user-item interaction information at the granularity of demands and modal interests.** V-pre and T-pre represent the user's interest in different modalities of the item.

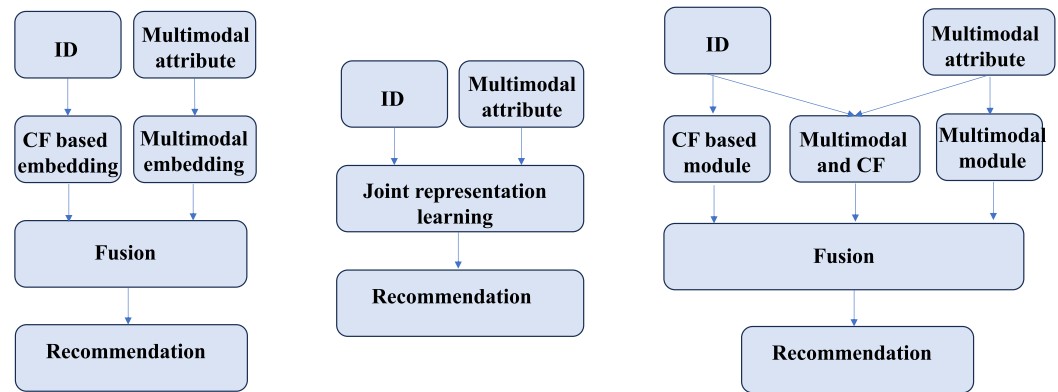

**Figure 2 An illustration of the cooperative relationship between two signals.**

from significant representation discrepancies between the two embedding spaces, leading to suboptimal fusion results. The second approach (Fig. 1, middle) jointly learns CF and MF signals within a unified framework (usually using graphs), enabling simultaneous optimization. While this method mitigates representation discrepancies, it risks conflicting optimization directions between CF and MF signals, potentially undermining overall performance. Both approaches struggle to adapt to diverse real-world scenarios, where user preferences may be driven by either collaborative signals, content similarity, or a combination of both. Current methods fails to consider these signals at a fine-grained level, thereby falling short in comprehensively modeling user preferences.

To address these limitations, we propose a novel Dual-path Selective Graph Recommender (DSGRec), which consists of four components including heterogeneous networks information collaboration (HNIC), behavior-aware modality signal

augmentation (BMSA), hypergraph-guided cooperative signal enhancement (HCSE) and adaptive fusion and prediction module.

To tackle the first limitation, HNIC divides interaction signals into two distinct types: collaborative signals (C-signals), dominated by similar users and obtained through collaborative filtering (CF), and preference signals (P-signals) guided by user interests and derived from MF. Specifically, the HNIC module performs independent message propagation on the user-item interaction graph and the multi-modal feature graph, enabling the separate capture of C-signals and P-signals. Additionally, the behavior-aware modality signal augmentation module adaptively fuses user preferences extracted from behavioral features in different environments with item modal features, thereby enriching P-signals. To further enhance the representation, the Hypergraph-guided cooperative signal enhancement module constructs global hyper-edges, supplementing information that traditional CF methods fail to capture due to hop distance limitations. This module learns the hybrid representation of C-signals and P-signals by incorporating global higher-order relationships. Finally, to address the second limitation, we introduce an adaptive fusion mechanism that dynamically adjusts the weights of P-signals, C-signals and the hybrid global feature of two signals based on their contributions in different scenarios as shown in Fig. 2 (right). By decoupling the learning processes while maintaining their interactions, our approach effectively captures the complexity of user-item interactions and overcomes the shortcomings of existing single-path or dual-path methods. The contribution of this work are summarized as follows:

1. We propose a novel division of interaction signals into two distinct types: Collaborative signals (C-signals), derived from CF, and P-signals derived from MF, enabling a more granular understanding of user-item interactions.

2. We design a fine-granularity learning framework that independently models CF-based embeddings, MF-based embeddings, and their fused representations, followed by an adaptive fusion of the three.

3. We conduct extensive experiments on three real-world datasets to validate the effectiveness of DSGRec. The evaluation results demonstrate that DSGRec has attained comparable performance with state-of-the-art methods.

The remainder of this article is organized as follows. "Related Work" reviews related work in graph-based recommendation and multi-modal recommendation. Then, "Problem Definition and Notations", "Methodology" and "Complexity Analysis" introduce the problem definition and proposed methodology. "Experiments" presents and analyzes the experimental results on three standard datasets. Finally, "Conclusion" concludes the work and discusses future research directions.

## RELATED WORK

### Multi-modal recommendation

With the growing prevalence of multimedia contents in the modern web era, many recommendation systems have evolved from single-modal approaches to multi-modal

methods that integrate various data types and features. VBPR (*He & McAuley, 2016*) considers the visual appearance of the items by extracting visual features from pre-trained network. Recently, many works introduce GNNs into multimodal recommendation. Besides multi-modal item representations, MMGCN (*Wei et al., 2019*) enhances user representations from modalitiy specific user-item interactions. MICRO (*Zhang et al., 2022a*) injects a multi-modal representation of item-item into items and captures the relationships between different items with graph convolution. Attention-guided multi-step fusion network (TMFUN) (*Zhou et al., 2023c*) and Self-supervised graph disentangled network (SGDN) (*Ren et al., 2023*) are both trained on the interaction data between users and items to get the deep relationship between users and items. Self-supervised interest transfer network (SITN) (*Sun et al., 2023*) and MBSSL (*Xu et al., 2023*) operate the nodes to obtain a better recommendation effect. SITN (*Sun et al., 2023*) aggregates nodes with semantic invariance in different semantic Spaces. Bias constrained contrastive learning (BCCL) (*Yang et al., 2023*) utilizes data augmentation with constrained biases to enhance sample quality. However, existing recommendation methods fail to consider attractiveness and demand as the bidirectional characteristics of user-item interactions, leading to limited recommendation efficacy within specific item categories. DSGRec adopts two different methods to capture attraction and demand signals on user-item interaction graphs, allowing for more sophisticated modeling of complex user behaviors. MGNM (*Mo et al., 2024*) effectively filters out noise from the modality features, ensuring the refined information is more closely aligned with user preferences. Counterfactual knowledge distillation (CKD) (*Zhang et al., 2024*) enhances the information utilization rate of each modality under single-modality guidance, effectively capturing complementary information between different modalities.

## Graph-based recommendation

In the field of recommender system, graph convolutional neural networks are favored by mainstream methods because of their excellent performance in high-order connectivity modeling. LightGCN (*He et al., 2020*) is a foundation stone for applying graph convolutional neural networks to multi-modal recommendation systems. Exploring user preferences from multiple modalities information of items (*Zhou et al., 2023a*; *Zhang et al., 2021*) has been a hot topic for MRS. For example, LATTICE (*Zhang et al., 2021*) enhances node representations in the graph by learning from multi-modal information. DRAGON (*Zhou et al., 2023a*) learns dual representations of users and items by constructing homogeneous and heterogeneous graphs. Multi-modal graph contrastive learning (MMGCL) (*Yi et al., 2022*) ensures the effective contribution of each modality through different data augmentation. FREEDOM (*Zhou & Shen, 2023*) freezes the large model and only trains the end of the model and graph denoising, achieving good results. Other methods such as DRAGON (*Xia et al., 2023*) learns dual representations of users and items by constructing both homogeneous and heterogeneous graphs. GraphCAR (*Xu et al., 2018*) uses graph convolutional network (GCN) to capture higher-order connectivity and enhance preference features, they have used GCN-based CF methods to explore higher-order relationships between users and items. Dynamic graph evolution learning (DGEL)

(*Tang et al., 2023*) uses the joint training of interaction matching task and prediction task to ensure the rapidness and timeliness of the recommendation system for the interaction between users and items. Multimodal graph meta contrastive learning (MGMC) (*Zhao & Wang, 2021*) assigns meta-learning to contrastive learning to provide generalization capabilities for graph contrastive learning. In constructing nodes, current methods use a single pathway to learn a signal representing the method; however, due to the influence of noise, the more accurate the signal learned in a single pathway, the more severe the phenomenon of overfitting in the experiment will be. DSGRec uses two pathways and four different methods to build nodes to improve the performance and enhance the robustness of the recommender system. Disentangled Graph Variational Auto-Encoder (DGVAE) (*Zhou & Miao, 2024*) leverages GCNs to encode and disentangle ratings and multimodal information, enabling it to learn latent representations of items from their neighboring items.

## Hypergraph learning for recommendation

The utilization of hyperedges connecting multiple nodes allows for the construction of hypergraphs, which can serve as a supplementary tool in CF for extracting unexplored user-item relationship information (*Gao et al., 2020*). The hypergraph neural networks (HGNN) (*Feng et al., 2019*) framework incorporates hyperedge convolution to manage data correlations during the representation learning process. A hardware-friendly recommendation algorithm based on hyperdimensional computing (HyperRec) (*Wang et al., 2020*) achieves more granular recommendation effects by stacking hypergraph convolution networks, residual gating layers, and fusion layers, Dual channel hypergraph collaborative filtering (DHCF) (*Ji et al., 2020*) learns user and item representations separately so that these two types of data can be interconnected while maintaining their specific attributes. Hypergraph Click-Through Rate prediction framework (HyperCTR) (*He et al., 2021*) leverages information interaction between users and micro-videos by considering different modalities, Multi-channel Hypergraph Convolutional Network (MHCN) (*Yu et al., 2021*) incorporates self-supervised learning to reduce aggregation loss through hierarchical mutual information maximization, dual channel hypergraph convolutional network (DHCN) (*Xia et al., 2021*) introduces a dual-channel hypergraph convolutional network and incorporates self-supervised tasks into network training. This approach enhances hypergraph modeling and improves the performance of recommendation tasks, Co-guided Heterogeneous Hypergraph Network (CoHHN) (*Zhang et al., 2022b*) employs a dual-channel aggregation mechanism within a heterogeneous hypergraph network to integrate diverse information from heterogeneous nodes and multiple relations. In the last couple of years, several research efforts have integrated hypergraphs with self-supervised learning techniques by employing self-supervised learning as a regularization method for hypergraph learning (*Xia, Huang & Zhang, 2022*). However, approaches like hypergraph contrastive collaborative filtering (HCCF) (*Xia et al., 2022*) and local and global graph learning for multimodal recommendation (lgmrec) (*Guo et al., 2024*) focus on self-supervised learning on hypergraph information. However, self-supervised learning without filtering information

similarly leads to the enhancement of noise, which undermines the ability of the model to capture item characteristics and user preferences. In contrast, our approach not only leverages hypergraphs as supplementary signals but also incorporates multi-modal features of items and user preferences, thereby enhancing the precision of recommendations.

## PROBLEM DEFINITION AND NOTATIONS

In this section, we introduce the notations used in our work and formally define the task within the overall framework. Let $\mathbf{U} = \{u\}$ denotes the set of users, and $\mathbf{I} = \{i\}$ represents the set of items. The input ID embeddings of user $u$ and item $i$ are denoted as $\mathbf{E}^{id} \in \mathbb{R}^{d \times (|I| + |U|)}$, where $d$ is the dimension of embedding vectors. For each item $i$, we represent its modality feature as $\mathbf{E}_i^m \in \mathbb{R}^{d_m \times |I|}$, where $d_m$ is the dimension of the feature, $m \in \mathbf{M}$ is the index of the modality, and $\mathbf{M}$ is the set of modalities. In this article, following previous work, we define $\mathbf{M} = \{v, t\}$, where $v$ and $t$ represent the visual and textual modalities, respectively. Additionally, the user's historical behavior data is denoted as $P \in \mathbb{R}^{|U| \times |I|}$, where each entry $P_{u,i} = 1$ if user $u$ has interacted with the item $i$; otherwise, $R_{u,i} = 0$. Naturally, the historical interaction data $P$ can be viewed as a sparse behavior graph $\mathcal{G} = \{\mathcal{V}, \mathcal{E}\}$, where $\mathcal{V} = \{\mathbf{U} \cup \mathbf{I}\}$ represents the set of nodes, and $\mathcal{E} = \{(u,i) | u \in \mathbf{U}, i \in \mathbf{I}, R_{u,i} = 1\}$ denotes the set of edges. The goal of multi-modal recommendation is to accurately predict the likelihood of interaction between a user and an item by ranking all items for the user based on the predicted scores $\hat{r}_{u,i}$:

$$\hat{r}_{u,i} = DSGRec(\mathbf{E}_{HNIC}, \mathbf{E}_f, \mathbf{E}_h), \tag{1}$$

where $DSGRec(\cdot)$ is our model that predicts the likelihood of user $u$ interacting with item $i$. Here, $\mathbf{E}_{HNIC}$ represents the embeddings obtained from Heterogeneous Network Information Collaboration (HNIC), which employs graph neural network to capture C-signals and P-signals on the user-item interaction graph with ID embedding and modality features, respectively. The $\mathbf{E}_f$ represents the embedding from Behavior-aware Modal Signal Augmentation, and the $\mathbf{E}_h$ represents the embeddings learned by Hypergraph-guided Cooperative Signal Enhancement module.

## METHODOLOGY

In this section, we present the detailed framework of our proposed method, DSGRec. As illustrated in Fig. 3, the framework consists of four main components: (1) heterogeneous networks based information collaboration (HNIC); (2) behavior-aware multi-modal signal augmentation (BMSE); (3) hypergraph-guided cooperative signal enhancement (HCSE); and (4) adaptive fusion and prediction.

### Heterogeneous networks-based information collaboration

The HNIC module serves as the foundational component, aiming to learn representations of user demand for items and the multi-modal preferences. These two parts are learned separately through the propagation of ID embeddings and multi-modal information on

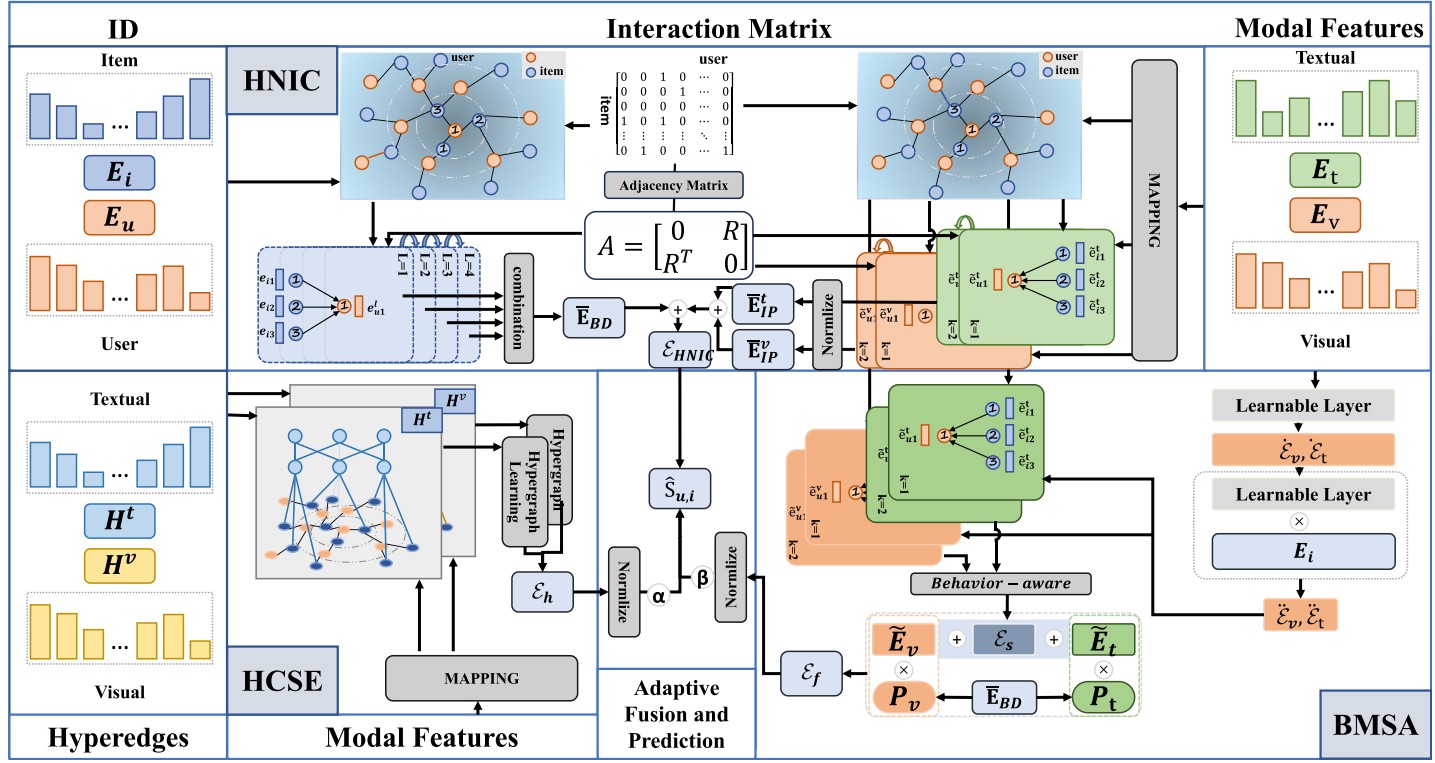

**Figure 3** The framework of the proposed DSGRec, which consists of four main modules.

the *User − Item* interaction graph, corresponding to behavior-oriented demand embedding and interest-guided preference embedding.

### Behavior-oriented demand embedding

We design a behavior-oriented demand embedding module to learn users' latent needs. To minimize the influence of different modality features on C-signals, we utilize the high-order interactions captured by message propagation on the *User − Item* embedding graph that only contains ID information. First, we construct a symmetric adjacency matrix **A** from the *User − Item* interaction matrix:

$$\mathbf{A} = \begin{bmatrix} 0 & P \\ P^\top & 0 \end{bmatrix}, \tag{2}$$

where $P$ is the user-item interaction matrix. We then employ LightGCN (*He et al., 2020*) to propagate the ID embeddings of users and items in the interaction graph, enabling the learning of high-order collaborative signals. The graph convolution at each layer can be formulated as:

$$\mathbf{E}^{id(l)} = \mathbf{E}^{id(l-1)} \mathbf{D}^{-\frac{1}{2}} \mathbf{A} \mathbf{D}^{-\frac{1}{2}}, \tag{3}$$

where $\mathbf{E}^{id(l)}$ is the embedding matrix at the $l$-th layer after graph convolution, and $\mathbf{E}^{id(0)}$ is the initial ID embeddings, and **D** is the diagonal degree matrix. To ensure fairness and

eliminate the difference in the number of interactions, we apply symmetric normalization $\frac{1}{\sqrt{N_u}\sqrt{N_i}}$ to the embedding, where $N_u$ is the number of items interacted with by user $u$, and $N_i$ is the number of users who interacted with item $i$. The representation at the $l$-th layer encodes information from $l$-order neighbors. After aggregating the information of high-order neighbors. After aggregating high-order neighbor information, we obtain the final embedding $\overline{\mathbf{E}}_{BD}$:

$$\mathbf{E}_{BD} = \frac{1}{L+1}\sum_{i=0}^{L}\mathbf{E}^{id(l)}. \tag{4}$$

### Interest-guided preference embedding

To identify which modality of an item attracts users, we propagate different item modalities on the graph to explore the most appealing information for users. Specifically, we first project the modality features of different dimensions, obtained from pre-trained models, into a unified embedding space $\mathbb{R}^d$.

$$\widetilde{\mathbf{E}}_i^m = \mathbf{E}_i^m \cdot \mathbf{W}_m, \tag{5}$$

where $\widetilde{\mathbf{E}_i^m}$ is the mapped multi-modal feature of items in $\mathbb{R}^{|I|\times d}$ of $i$. $\mathbf{W}_m$ is a learnable transformation matrix. Then, we aggregate the multi-modal information of the neighbor set on the $U - I$ graph to represent user $u$'s preference for each modality:

$$\tilde{e}_u^m = \frac{1}{|N_u|}\sum_{i\in N_u}\tilde{e}_i^m, \tag{6}$$

where $\tilde{e}_u^m$ is user $u$'s preference feature for the $m$-th modality, and $N_u$ represents the neighbor set of user $u \in U$ on the user-item interaction graph $\mathbf{G}$. This operation ensures that user's needs and items appeal are modeled independently. The message propagation at the $l$-th graph convolution layer is formulated as:

$$\mathbf{E}^{m(l)} = \mathbf{E}^{m(l-1)}\mathbf{D}^{-\frac{1}{2}}\mathbf{A}\mathbf{D}^{-\frac{1}{2}}, \tag{7}$$

where $\mathbf{E}^{m(0)} = [e_{u_1}, \ldots, e_{u_{|U|}}, e_{i_1}, \ldots, e_{i_{|I|}}] \in \mathbb{R}^{(|U|+|I|)\times d}$ is the multi-modal feature matrix of items and users. Finally, we take the last layer's output as the final high-order modality embedding:

$$\overline{\mathbf{E}}_{IP}^m = \mathbf{E}^{m(L)}, \tag{8}$$

where $L$ is the number of layers.

### Dual-path embeddings fusion

After learning the demand embedding and preference embedding separately, we merge the two embeddings to obtain a fusion feature $\mathbf{E}_{HNIC}$:

$$\mathbf{E}_{HNIC} = \mathbf{E}_{BD} + \sum_{m\in M}\text{NORM}(\mathbf{E}_{IP}^m). \tag{9}$$

where *NORM* is a normalization function to alleviate the value scale difference among ID embeddin and preference embeddings.

## Behavior-aware multi-modal signal augmentation

Aiming to make the learned multi-modal feature complement with each other, we design a behavior-aware multi-modal signal augmentation module. Specifically, we first transform the original modality embeddings into shared feature space through a linear transform:

$$\ddot{\mathbf{E}}_i^m = \mathbf{W}_1 \mathbf{E}_i^m + b_1, \tag{10}$$

where matrix $\mathbf{W}_1 \in \mathbb{R}^{d \times d_m}$ and the bias vector $b_1 \in \mathbb{R}^d$ in the transition here is trainable, unlike $\mathbf{W}_m$. Then, a behavior-guided purifier is utilized to select the preference-relevant modality features $\ddot{\mathbf{E}}_i^m$ from the item's representation $\mathbf{E}_i^{id}$, which is proven to be effective in previous work (MGCN; *Yu et al., 2023*). The behavior-guided purifier $f_{gate}^m$ is defined as follow:

$$\ddot{\mathbf{E}}_i^m = f_{gate}^m(\mathbf{E}_i^{id}, \dot{\mathbf{E}}_i^m) = \mathbf{E}_i^{id} \odot \sigma(\mathbf{W}_2(\dot{\mathbf{E}}_i^m + b_2)), \tag{11}$$

where matrix $\mathbf{W}_2 \in \mathbb{R}^{d \times d_m}$ and $b_2$ are learnable parameters, $\odot$ represents the element-wise product, and $\sigma$ is the sigmoid function.

To obtain an accurate item modality vector and avoid contamination from user preferences, we only propagate modality features in the $I - I$ graph. First, we conduct KNN sparsification (*Chen, Fang & Saad, 2009*) on the $I - I$ graph. KNN sparsification can effectively reduce training costs and remove unnecessary noise as much as possible. The similarity between item $a$ and $b$ on modality $m$ is denoted as $s_{a,b}^m$:

$$s_{a,b}^m = \frac{(e_a^m)^T e_b^m}{||e_a^m|| \, ||e_b^m||}, \tag{12}$$

where $e_a^m$ and $e_b^m$ represent the feature of item $a$ and $b$ in modality $m$. To ensure that the edges retained in the graph are truly effective, we retain the edges with the top $k$ highest similarity in each row and remove the other edges. All the similarity values construct a affinity matrix $\mathbf{S}^m$ in Eq. (13), where the element in the row $a$ and column $b$ of it is computed with Eq. (12).

$$\dot{\mathbf{S}}^m = \begin{cases} s_{a,b}^m, & s_{a,b}^m \in \text{top} - \text{K}(s_{a,c}, c \in \mathbf{I}), \\ 0, & \text{otherwise.} \end{cases} \tag{13}$$

We also normalize the item-item affinity matrix to prevent gradient explosion, which has been proven effective in prior work (*Chen, Fang & Saad, 2009*). The normalizeditem-item affinity matrix is defined as:

$$\ddot{\mathbf{S}}^m = \mathbf{D}^{m\frac{1}{2}} \dot{\mathbf{S}}^m \mathbf{D}^{m-\frac{1}{2}}. \tag{14}$$

Here, $\mathbf{D}^m$ is the diagonal matrix of $\dot{\mathbf{S}}^m$. Then, we propagate all item modal features $\ddot{\mathbf{E}}_i^m$ through the corresponding $Item - Item$ affinity matrix $\dot{\mathbf{S}}^m$ by the LightGCN (*He et al., 2020*):

$$\overline{\mathbf{E}}_i^m = \dot{\mathbf{S}}^m \ddot{\mathbf{E}}_i^m. \tag{15}$$

It can enrich features by capturing the common characteristics of similar items. However, in the $I - I$ view, as the propagation path increases, the semantic similarity of the node modality features significantly decreases. Stacking multiple graph convolution layers not only leads to the over-smoothing problem of nodes but also easily captures noise features. Finally, we obtain the second user modality $\overline{\mathbf{E}}_u^m$ under the $I - I$ graph, user $u$'s modality feature $\bar{e}_u^m$ is expressed as:

$$\bar{e}_u^m = \sum_{i \in N_u} \frac{1}{\sqrt{|N_i||N_u|}} \bar{e}_i^m, \tag{16}$$

where $\bar{e}_i^m$ is the vector of $\overline{\mathbf{E}}_i^m$. By concatenating $\overline{\mathbf{E}}_u^m$ with $\overline{\mathbf{E}}_i^m$, we obtain the behavior-related modal aggregation auxiliary feature $\overline{\mathbf{E}}^m \in \mathbb{R}^{d \times (|U|+|I|)}$ for each modality. To model the multi-modal preference signals with a fine-grained fusion mechanism, we design a behavior-aware fusion $P_m$ for each modality $m$. This module learns and adjusts parameters to determine the weight of each item's attraction to users based on the behavior information matrix. It is defined as:

$$\mathbf{P}_m = \sigma(\mathbf{W}_3 \mathbf{E}_{BD} + b_3), \tag{17}$$

where $\mathbf{W}_3 \in \mathbb{R}^{d \times d_m}$ and $b_3 \in \mathbb{R}^d$ are learnable parameters, and $\sigma$ is the sigmoid function. $P_m$ describe the importance of each modality for different users. For flexible fusion weight allocation based on user modality preferences, we decompose the modal features into common features and unique features. First, we assume that the common features, derived from interaction information, are highly similar to the modal features of the user's target needs. Then, we capture these common features (*Vaswani et al., 2017*; *Wang, Wu & Hoashi, 2019*) using an attention mechanism, resulting in the modal weight matrix $\alpha$:

$$\alpha_m = \text{softmax}(q_1^\top \tanh(\mathbf{W}_4 \mathbf{E}^m + b_4)), \tag{18}$$

where $q_1 \in \mathbb{R}^d$ is the attention vector, $\mathbf{W}_4 \in \mathbb{R}^{d \times d_m}$ and $b_4 \in \mathbb{R}^d$ are the weight matrix and bias vector, respectively. These parameters are shared across all modalities. The shared modal features $\mathbf{E}_s$ are obtained by summing the weighted modality features:

$$\mathbf{E}_s = \sum_{m \in M} a_m \overline{\mathbf{E}}_m. \tag{19}$$

The unique modal features $\widetilde{\mathbf{E}}_m$ are then derived by subtracting the shared features $\mathbf{E}_s$:

$$\widetilde{\mathbf{E}}_m = \overline{\mathbf{E}}_m - \mathbf{E}_s. \tag{20}$$
Finally, we adaptively fuse the modal-specific features $\widetilde{\mathbf{E}}_m$ with the shared features $\mathbf{E}_s$ to obtain the behavior-related modal aggregation auxiliary embeddings $\mathbf{E}_f$:

$$\mathbf{E}_f = \mathbf{E}_s + \frac{1}{|M|} \sum_{m \in M} \widetilde{\mathbf{E}}_m \odot \mathbf{P}_m. \tag{21}$$

To ensure that the auxiliary information $\mathbf{E}_f$ is learned effectively under sparse data conditions and to better explore the relationship between CF signals and MF signals, we design a self-supervised auxiliary task to optimize $\mathbf{E}_f$ with the loss $\mathbf{L}_f$:

$$\mathbf{L}_f = \sum_{i \in I} -\log \frac{\exp(e_{i,f} \cdot e_{i,id}/\tau_1)}{\sum_{m \in I} \exp(e_{m,f} \cdot e_{m,id}/\tau_1)} + \sum_{u \in U} -\log \frac{\exp(e_{u,f} \cdot e_{u,id}/\tau_1)}{\sum_{n \in U} \exp(e_{n,f} \cdot e_{n,id}/\tau_1)}. \tag{22}$$

Here, $L_f$ can learn meaningful feature representations by maximizing the similarity of positive sample pairs while minimizing the similarity of negative sample pairs, and $\tau_1$ is a temperature hyper-parameter, $e_{u,f}$ and $e_{i,f}$ denote the features of $u$ and $i$ in $\mathbf{E}_f$, $e_{u,id}$ and $e_{i,id}$ denote the features of $u$ and $i$ in $\overline{\mathbf{E}}_{BD}$.

## Hypergraph-guided cooperative signal enhancement

The over-smoothing effect in deeper graph-based CF architectures can lead to indistinguishable user representations and a degradation in recommendation quality. Inspired by the ability of hyper-graphs to capture complex high-order relationships globally, we introduce hyper-graph to capture high-order global hybrid representations of CF signal and MF signal missed that are missed in local learning due to data sparsity.

To make the model to learn the hypergraph structure adaptively, we define learnable vectors $\mathbf{V}_k^m \in \mathbb{R}^d$ to represent a set of hyper-edges, where $k \in K$ and $K$ is the number of hyper-edges, $m \in M$ and $M$ represents the set of modalities. As illustrated in Fig. 4, the process include hyperedge learning, matrix calculation and hyper-graph embedding learning.

First, we learn the hyperedges of item nodes from the original modality embeddings in the low-dimensional embedding space:

$$\mathbf{H}_i^m = \widetilde{\mathbf{E}}_i^m \cdot \mathbf{V}^{m\top}, \tag{23}$$

where $\mathbf{H}_i^m \in \mathbb{R}^{d \times |K|}$ represents the item hyperedge dependency matrices. $\widetilde{\mathbf{E}}_i^m$ is the original item modality feature matrix computed with Eq. (5), and $\mathbf{V}_m = [V_1^m, .., V_K^m] \in R^{K \times d_m}$ is the hyperedge vector matrix. Based on $\mathbf{H}_i^m$ and user behavior, the hyperedge $\mathbf{H}_u$ for users can be obtained as follows:

$$\mathbf{H}_u^m = \mathbf{A}_u \cdot \mathbf{H}_i^{m\top}, \tag{24}$$

where $\mathbf{H}_u^m$ represents user-hyperedge dependency matrices, and $\mathbf{A}_u \in \mathbb{R}^{|U| \times |I|}$ is the user-related adjacency matrix extracted from adjacency matrix $\mathbf{A}$. This ensures that the hyperedges connect nodes with similar modalality features, allowing the hyperedge embedding to globally correct the embedding vector previously obtained by CF through
# Hyper-edges Learning    Dependence Matrix    Embedding Propagation

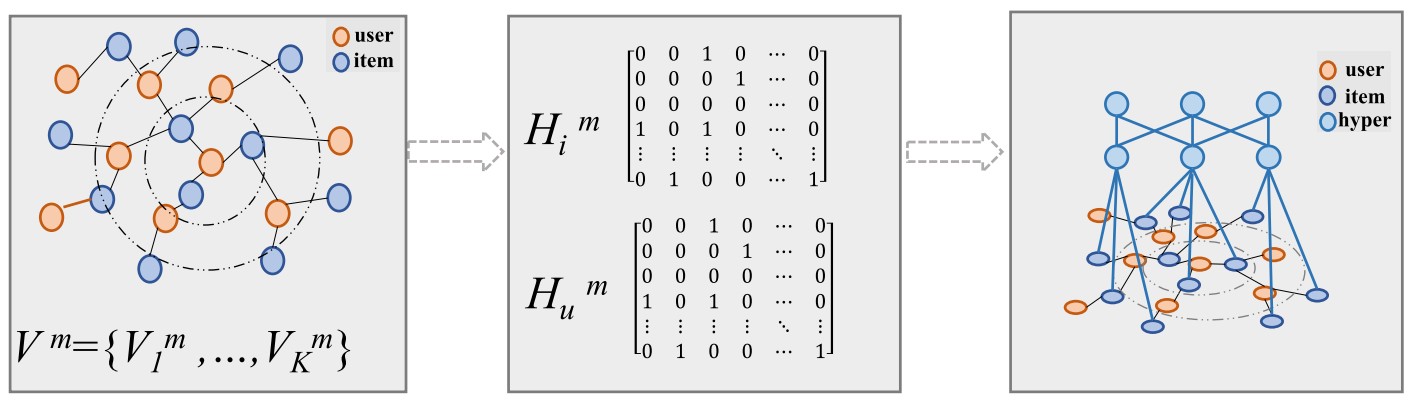

**Figure 4** Construction of hypergraphs for global information extraction.

user behavior information. To ensure fairness in each node's contribution to the hypergraph and avoid the same node being captured by multiple hyperedges, we apply Gumbel-Softmax reparameterization (*Jang, Gu & Poole, 2017*) for each node:

$$\tilde{h}_{i,*}^m = \text{SOFTMAX}\left(\frac{\log\delta - \log(1-\delta) + h_{i,*}^m}{\tau}\right), \tag{25}$$

where $\tilde{h}_{i,*}^m \in \mathbb{R}^K$ is the $i$-th row vector of $\mathbf{H}_i^m$. $\delta \in \mathbb{R}^K$ is a noise vector, where each value $\delta_k \sim \text{Uniform}(0,1)$, and $\tau$ is the temperature hyperparameter. The SOFTMAX represents the softmax function, which ensures differentiable smapling. Subsequently, we obtain the enhanced item-attribute hypergraph dependency matrix $\widehat{\mathbf{H}}_i^m$. By performing a similar operation on $\mathbf{H}_u^m$, we derive the enhanced user-attribute relationship matrix $\widetilde{\mathbf{H}}_u^m$. Similar with traditional CF approaches, we use hyperedges as bridges for the message-passing mechanism. Due to the characteristics of hyperedges, modality information is no longer limited by hop distance during transmission. The information of the entire graph can be propagated to each user $u$ and item $i$ using the following formulas:

$$\mathbf{E}_i^{m(l+1)} = SH_i \cdot \mathbf{E}_i^{m(l)}, \mathbf{E}_u^{m(l+1)} = SH_u \cdot \mathbf{E}_u^{m(l)}. \tag{26}$$

where $\mathbf{E}_i^{m(l)}$ and $\mathbf{E}_u^{m(l)}$ represents the hypergraph embedding matrices of items and users under modality $m$ at the $l$-th layer, respectively. We use $\mathbf{E}_{BD}$ as the initial embedding when $l = 0$. $SH_i$ and $SH_u$ are matrices describing the global relationship between nodes with learned hyperedges, computed as follows based on prior work (*Guo et al., 2024*):

$$SH_i = Drop(\widetilde{\mathbf{H}}_i^m) \cdot Drop(\widetilde{\mathbf{H}}_i^{m\top}), SH_u = Drop(\widetilde{\mathbf{H}}_u^m) \cdot Drop(\widetilde{\mathbf{H}}_i^{m\top}) \tag{27}$$

where $Drop()$ represents the dropout function.

In hypergraph transmission, by using the hypergraph dependency perceived through user behavior as input, we achieve global CF and MF information propagation, supplementing or denoising the information missed in the local information transmission

of previous work. Subsequently, the embeddings of users and items are stacked to obtain the hybrid feature embedding matrix $\mathbf{E}_h$:

$$\mathbf{E}_h = \sum_{m \in M} Concat(\mathbf{E}_u^{m(l)}, \mathbf{E}_i^{m(l)}), \tag{28}$$

where $\mathbf{E}_u^{m(l)} \in \mathbb{R}^{|U| \times d}$ and $\mathbf{E}_i^{m(l)} \in \mathbb{R}^{|I| \times d}$ are the embedding matrices representing the global hybrid features of user $u$ and item $i$ under modality $m$ at the $l$-th layer, respectively. $Concat()$ represents the concatenate operation.

To explore the collaborative relationship between demand signals and modality embedding signals, we optimize each signal independently before fusion. For the HCSE module, due to data scarcity and the complexity of comparing cross-modal information, we utilize contrastive learning to optimize $E_h$ in a self-supervised manner (*Gutmann & Hyvärinen, 2010*). Specifically, we treat embeddings of the same user/item under the different modalities as postive pairs and embeddings of different users/items under different modalities as negative pairs:

$$L_h = \sum_{u \in U} -\log \frac{\exp(\mathbf{e}_u^{v(l)} \cdot \mathbf{e}_u^{t(l)} / \tau_2)}{\sum_{u' \in U} \exp(\mathbf{e}_u^{v(l)} \cdot \mathbf{e}_{u'}^{t(l)} / \tau_2)} + \sum_{i \in I} -\log \frac{\exp(\mathbf{e}_i^{v(l)} \cdot \mathbf{e}_i^{t(l)} / \tau_2)}{\sum_{i' \in I} \exp(\mathbf{e}_i^{v(l)} \cdot \mathbf{e}_{i'}^{t(l)} / \tau_2)}, \tag{29}$$

where $\tau_2$ is the temperature factor for this loss function, and $u'$ and $i'$ are randomly sampled negative samples for user $u$ and item $i$, respectively.

## Adaptive fusion and prediction

We obtain the final embedding $e$ of users and items by aggregating the collaborative embedding $\mathbf{E}_{HNIC}$ from the heterogeneous network, the augmented multi-modal features $\mathbf{E}_f$, and the global hybrid features $\mathbf{E}_h$ from the hyper-graph network:

$$e = \mathbf{E}_{HNIC} + \alpha \cdot NORM(\mathbf{E}_h) + \beta \cdot NORM(\mathbf{E}_f), \tag{30}$$

where $NORM()$ is a normalization function to alleviate the value scale difference among embeddings, and $\alpha$ and $\beta$ are weighting factors. Following *He et al. (2020)*, we adopt the inner product to calculate the prediction score between user $u$ and item $i$.

$$\widehat{\mathbf{r}}_{u,i} = e_u^T e_i. \tag{31}$$

After obtaining the $\widehat{\mathbf{r}}_{u,i}$, we use Bayesian personalized ranking (BPR) loss (*Rendle et al., 2012*) to optimize the parameters of DSGRec:

$$\mathbf{L}_{BPR} = -\frac{1}{|D|} \sum_{(u,i^+,i^-) \in D} \ln \sigma(\widehat{\mathbf{r}}_{u,i^+} - \widehat{\mathbf{r}}_{u,i^-}), \tag{32}$$

where $(u, i^+, i^-)$ is a set of triples for training. Here, $u$ is the user embedding vector, $i^+$ is an item that user $u$ has interacted with, and $i^-$ is a randomly sampled negative item from the dataset. Finally, we integrate the loss functions of each component as follows:

$$\mathbf{L} = \mathbf{L}_{BPR} + \lambda_f \mathbf{L}_f + +\lambda_h \mathbf{L}_h + \lambda_E ||\phi||_2, \tag{33}$$

where $\lambda_f$, $\lambda_h$, and $\lambda_E$ are hyperparameters for weighting the loss terms. $\phi$ represents the model parameters, which are regularized using $L_2$ regularization to prevent overfitting.

## COMPLEXITY ANALYSIS

According to the architecture of DSGRec, we elaborate on each component to analyze the time cost. For the heterogeneous networks information collaboration module, the behavior-oriented demand embedding has computational complexity $\mathcal{O}(L * |E| * d)$, where $L$ is number of layers for graph convolution, and $E$ represents the number of interactions recorded in the user-item interaction graph, $d$ is the embedding size, which we set to 64. The interest-guided preference embedding module embedding complexity is $\mathcal{O}(|M| * |E| * d_m)$, where $M$ represents the number of modalities, $d_m$ is the dimension of the feature. The overall time complexity of the heterogeneous networks information collaboration module is $\mathcal{O}((L * d + |M| * d_m) * |E|)$.

For the behavior-aware modal signal augmentation, the computation cost primarily comes from the KNN algorithm with $O(|I|^2 * d_m)$, and the cost of the contrastive learning $\mathcal{O}(b * (|U| * d + |I| * d_m))$. Here, $b$ is the batch size, $|U|$ and $|I|$ represent the number of user nodes and item nodes, respectively.

For the hypergraph-guided cooperative signal enhancement, the time complexity of hypergraph dependency construction is $\mathcal{O}(|M| * K * |I| * (|U| + d_m))$, where $K$ represents the number of hyperedges. The message passing schema has a time complexity of $\mathcal{O}(|M| * (|I| * H + |U|) * K * ES)$, where $H$ represents the number of hypergraph layers and $ES$ is the embedding size. The cost for contrastive learning is $\mathcal{O}(b * (|U| + |I|) * ES)$, which is same as behavior-aware modal signal augmentation module.

## EXPERIMENTS

In this section, we conduct extensive experiments on three datasets to evaluate the effectiveness of DSGRec and address the following research questions through the experimental results:

- **RQ1:** Can DSGRec outperform state-of-the-art baseline models of different types in terms of recommendation performance?
- **RQ2:** What is the contribution of each components to the overall method?
- **RQ3:** How do different hyperparameter settings influence the performance of DSGRec?

### Experimental setup
#### *Dataset*
To validate our model, we conduct comprehensive experiments on three widely used Amazon datasets (*McAuley et al., 2015*): (a) Baby, (b) Sports and Outdoors, and (c) Clothing Shoes, which we refer to as Baby, Sports, and Clothing in brevity (https://nijianmo.github.io/amazon/index.html). These datasets contain both textual and visual modality information, making them suitable for evaluating multi-modal models. Following previous work (*Guo et al., 2024*), we utilize 4,096-dimensional visual features and 384-dimensional text features provided by the dataset. Table 1 summarizes the

**Table 1 Statistics of the three experimental datasets.**

| Dataset | User | Item | Behavior | Sparsity |
|---|---|---|---|---|
| Baby | 19,445 | 7,050 | 160,792 | 99.883% |
| Sports | 35,598 | 18,357 | 296,337 | 99.955% |
| Clothing | 39,387 | 23,033 | 278,677 | 99.969% |

statistics of these datasets, including the number of users, items, interactions, and the degree of sparsity.

### Experimental settings

We randomly split the user-item interaction data into training, validation and testing in a ratio of 8:1:1. We adopt two widely used metrics: Recall (R@n) and Normalized Discounted Cumulative Gain (NDCG@n) (*He et al., 2015*).

We set the default batch size, learning rate, and embedding size to 2,048, 0.001, and 64, respectively. The optimal hyperparameters are determined through grid search on the validation set. Specifically, we tune the number of graph propagation layers in $\{1, 2, 3, 4\}$, the number of hyperedges $K$ in $\{1, 2, 4, 8, 16, 32, 64, 128, 256\}$, the factors $\alpha$ and $\beta$ in $\{-1, -0.9, \ldots, 1.0\}$, and the dropout rate $p$ in $\{0.1, 0.2, \ldots, 1.0\}$. The loss weighting $\lambda_1$, $\lambda_2$, and $\lambda_3$ are searched in $\{1e-6, 1e-5, \ldots, 0.1\}$. For the contrastive learning auxiliary tasks, we set the temperature coefficient $\tau_1 = 0.2$, $\tau_2 = 0.2$. We employ an early stopping mechanism during the training process based on R@20 on the validation set.

The optimization process employees the stochastic gradient descent (SGD) algorithm. In implementation, the Adam optimizer is used to adjust the learning rate.

### Baselines

To evaluate the effectiveness of DSGRec, we compare it with two types of representative baseline models, including:

(1) CF-based models focusing on interaction signals: BPR (*Rendle et al., 2012*), LightGCN (*He et al., 2020*), HCCF (*Xia et al., 2022*), and LGMRec (*Guo et al., 2024*);

(2) Multi-modal based models using multi-modal signals as side information: VBPR (*He & McAuley, 2016*), MMGCN (*Wei et al., 2019*), MICRO (*Zhang et al., 2022a*), BM3 (*Zhou et al., 2023b*), FREEDOM (*Xia et al., 2022*), MGCN (*Yu et al., 2023*).

### Overall performance comparison (RQ1)

Table 2 presents the performance of all methods on three datasets. Specifically, our method outperforms baseline models and shows strong competitiveness against the latest models on two datasets, which improved by 2.84%, and 2.17% in terms of Recall@20 for the baby and sports, respectively. In addition, the performance on the Clothing dataset is superior to most baseline method. Further analysis of the two types of models reveals: (1) Comparison with CF-type models: Our model performs better on the Sports and Clothing datasets, which we attribute to its emphasis on modality information. Some CF models overestimate

**Table 2 The overall performance of DSGRec and different types of recommendation models on three datasets.** We use bold to mark the best results and underline to mark the second best.

| Datasets | Baby | | Sports | | Clothing | |
|---|---|---|---|---|---|---|
| Metrics | R@20 | N@20 | R@20 | N@20 | R@20 | N@20 |
| BPR(UAI '09) | 0.0607 | 0.0261 | 0.0690 | 0.0314 | 0.0315 | 0.0144 |
| LightGCN(SIGIR '20) | 0.0732 | 0.0320 | 0.0829 | 0.0379 | 0.0514 | 0.0227 |
| HCCF(SIGIR '22) | 0.0756 | 0.0332 | 0.0857 | 0.0394 | 0.0533 | 0.0235 |
| LGMRec(AAAI '24) | 0.0986 | 0.0436 | 0.1068 | 0.0480 | 0.0828 | 0.0371 |
| DGVAE(IEEE '24) | 0.1009 | 0.0436 | 0.1123 | **0.0506** | 0.0917 | 0.0412 |
| VBPR(AAAI '16) | 0.0663 | 0.0284 | 0.0854 | 0.0378 | 0.0412 | 0.0191 |
| MMGCN(MM'19) | 0.0749 | 0.0315 | 0.0825 | 0.0382 | 0.0564 | 0.0253 |
| MICRO(IEEE '23) | 0.0905 | 0.0406 | 0.1026 | 0.0463 | 0.0743 | 0.0332 |
| BM3(WWW'23) | 0.0857 | 0.0378 | 0.0979 | 0.0437 | 0.0669 | 0.0295 |
| MGCN(MM '23) | 0.0964 | 0.0427 | 0.1106 | 0.0496 | **0.0945** | **0.0428** |
| CDK(ACM'24) | 0.0866 | 0.0389 | 0.1004 | 0.0463 | 0.0770 | 0.0363 |
| **DSGRec** | **0.1014** | **0.0438** | **0.1130** | 0.0500 | 0.0891 | 0.0406 |

**Table 3 Ablation analysis of DSGRec variants with performance benchmarking.** Best results are highlighted in bold across all variants.

| Datasets | Baby | | Sports | | Clothing | |
|---|---|---|---|---|---|---|
| Metrics | R@20 | N@20 | R@20 | N@20 | R@20 | N@20 |
| w/o BDE | 0.0943 | 0.0414 | 0.0337 | 0.0860 | 0.0847 | 0.0376 |
| w/o IPE | 0.0789 | 0.0377 | 0.0493 | 0.0210 | 0.0702 | 0.0314 |
| w/o BMSA | 0.0989 | 0.0436 | 0.1119 | 0.0491 | 0.0871 | 0.0399 |
| w/o HCSE | 0.0987 | 0.0433 | 0.1123 | 0.0497 | 0.0808 | 0.0364 |
| DSGRec | 0.1014 | 0.0438 | 0.1130 | 0.0500 | 0.0891 | 0.0406 |

the role of collaborative signals in capturing user interests, leading to suboptimal performance in interest-dominated environments. (2) Comparison with multi-modal models: Our model's performance on the demand-guided baby dataset indicates that some models overly rely on modal information learning while neglecting demand information exploration. DSGRec adaptively integrates high-order modality information with demand signals, ensuring robust performance.

## Ablation study (RQ2)

To investigate the complexity of user behavior and explore the specific contributions of different modules, we conduct an ablation study by comparing DSGRec with four variants:

- w/o BDE: Disable the behavior-oriented demand embedding module but retain its participate in auxiliary tasks.

| Table 4 Parameter settings. | | | |
|---|---|---|---|
| **Dataset** | **Baby** | **Sports** | **Clothing** |
| $\lambda_f$ | 0.000001 | 0 | 0.000001 |
| $\lambda_h$ | 0.0001 | 0.001 | 0.0001 |
| $\lambda_E$ | 0.001 | 0.001 | 0.001 |
| $\alpha$ | 0.6 | 0.2 | −0.9 |
| $\beta$ | −0.3 | −0.3 | 0.8 |
| Hyperedges | 4 | 4 | 64 |
| $\tau$ | 0.5 | 0.2 | 0.2 |
| $\tau_1$ | 0.2 | 0.2 | 0.2 |
| $\tau_2$ | 0.5 | 0.2 | 0.2 |
| $\mathbf{E}_{id} layer$ | 2 | 4 | 2 |
| $\overline{\mathbf{E}}_{IP}^m layer$ | 2 | 3 | 3 |
| Hypergraph layer | 1 | 4 | 2 |

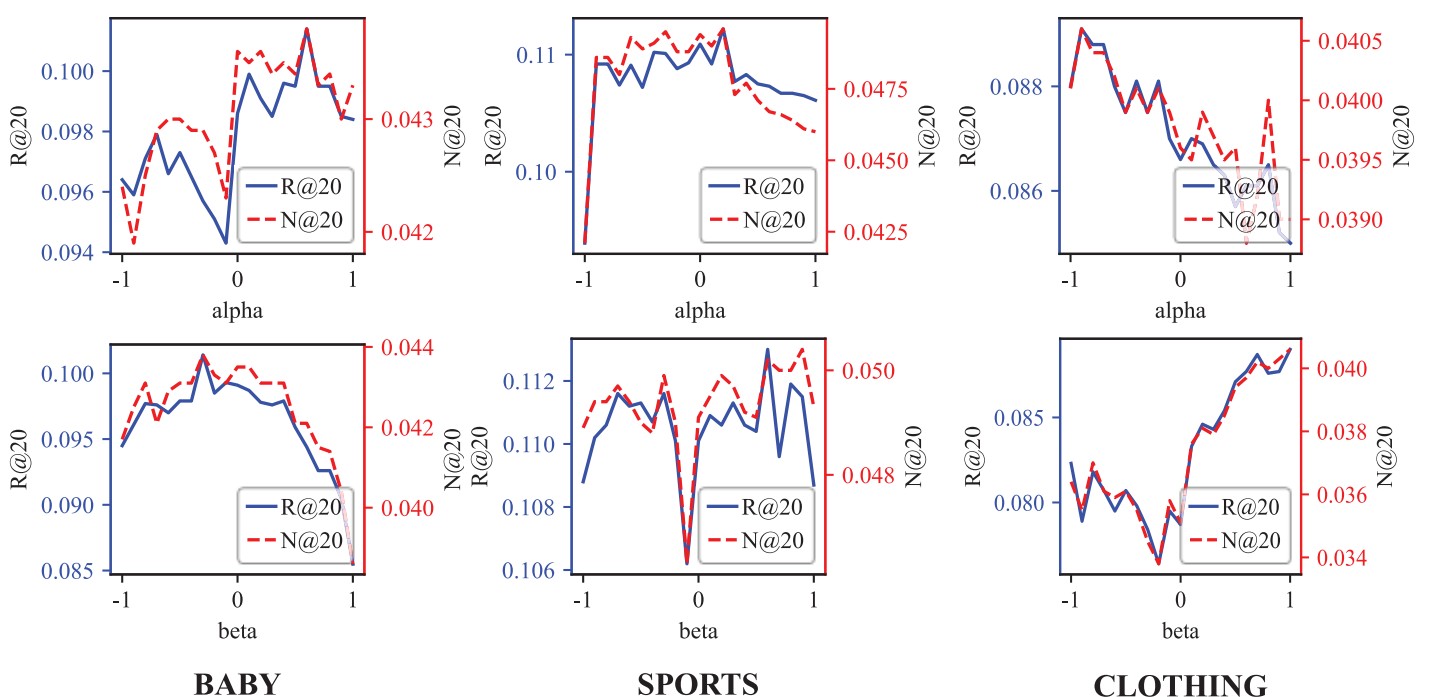

**Figure 5 The Recall@20 and NDCG@20 results of different $\alpha$ and $\beta$ weight and vision weight.**

- w/o IPE: Remove the interest-guided preference embedding module, and set $\mathbf{E}_{HNIC} = \overline{\mathbf{E}}_{BD}$.
- w/o BMSA: Remove the behavior-aware modal signal augmentation module.
- w/o HCSE: Remove the hypergraph-guided cooperative signal enhancement module.

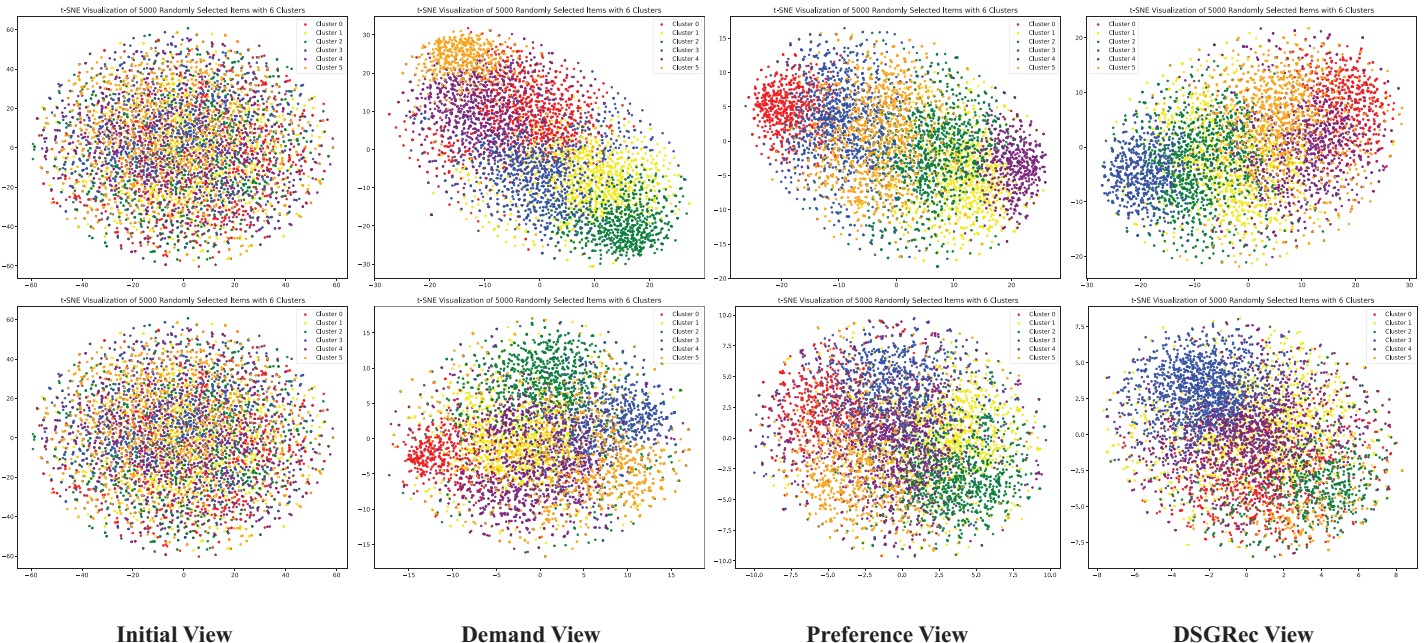

|  |  |  |  |
|---|---|---|---|
| **Initial View** | **Demand View** | **Preference View** | **DSGRec View** |

**Figure 6** **The 2D distributions of the learned demand and preference pathways in DSGRec.** Using t-SNE visualization, we can intuitively see that DSGRec effectively retains the independent semantics of the dual representations and can adaptively fuse information across pathways.

Table 3 reports Recall@20 and NDCG@20 of these variants on the three datasets, leading to the following findings:

(1) Among these four models, the variants w/o BDE and w/o IPE exhibit the worst performance, demonstrating that user behavior cannot be comprehensively modeled by methods using a single signal. DSGRec's decomposition of behavioral signals into demand and perefernce components enables independent learning and effective collaboration.

(2) The modules w/o BMSA and w/o HCSE indeed play a role in optimization. The performance gap varies across datasets, indicating that interest and demand signals are not conflicting and can sometimes be optimized by a single module. This highlights the limitations of single-path modeling and the benefits of multi-module collaboration.

## Hyperparameter discussion (RQ3)

In this section, we discuss the hyperparameters settings for the three datasets in our experiments. Table 4 details the optimal parameter settings. Additionally, Fig. 5 presents the Recall@20 and NDCG@20 for different $\alpha$ and $\beta$ in detail.

As the Table 4 demonstrates, the weight $\alpha$ of Behavior-aware Modal Signal Augmentation is positive for the Baby and Sports datasets but negative for the Clothing datasets. Conversely, the weight $\beta$ of Hypergraph-guided Cooperative Signal Enhancement is opposite to $\alpha$. This pattern, illustrated in Fig. 5, aligns with our hypothetical model: the interpretation of interactive behavior should be context-dependent.

In the demand-driven Baby dataset, modal information may interfere with the recommendation accuracy. On the contrary, due to the GCN limitations, the demand

signal requires hypergraph enhancement to achieve optimal performance. In the Clothing dataset, the optimal α value is −0.9, indicating that the demand signal struggles to capture user preferences effectively. The preference signal demonstrates a significant advantage, explaining why DSGRec underperforms compared to multi-modal methods on this dataset.

### Visualization

We utilize a visualization module to create 2D graphs showing the representation of nodes from initial view, demand view, preference view and DSGRec view. We randomly sample 5,000 items from the dataset and map their embedding vectors into a 2-dimensional space using t-SNE (*Van der Maaten & Hinton, 2008*). In Fig. 6, the first row visualizes initial embedding vectors, and the second row shows results after training. Each column corresponds to different method variants, as in the ablation study. Node colors represent different clusters. From the figure, we observe that the initial view embeddings displays a random distribution of users and items, while other views result in more distinguishable distributions.

## CONCLUSION

In this article, we propose DSGRec, a novel dual-path recommendation framework designed to model user-item interactions more effectively. DSGRec decomposes explicit interaction signals into two types of implicit interaction information: demand signals and preference signals. Subsequently, an adaptive fusion mechanism is then employed to facilitate fine-grained collaboration between these signals, significantly improving the model's robustness and recommendation performance. Extensive experiments on three real-world datasets demonstrate the dual-path representation could effectively sutiable to different interaction context.

Despite its advantages, DSGRec's dual-channel architecture introduces more hyperparameters and result in longer training times. In future work, we will focus on designing more effective and efficient fusion architecture for integrating collaborative filtering signals and multi-modal signals.

### Funding
The authors received no funding for this work.

### Competing Interests
The authors declare that they have no competing interests.

### Author Contributions
- Zihao Liu conceived and designed the experiments, performed the experiments, analyzed the data, performed the computation work, prepared figures and/or tables, and approved the final draft.

- Wen Qu conceived and designed the experiments, analyzed the data, prepared figures and/or tables, authored or reviewed drafts of the article, and approved the final draft.

## Data Availability

The source code of the method is available in the Supplemental File and at GitHub: https://github.com/gustab77/DSGRec.

The Amazon Review Data (2018) is available at: https://nijianmo.github.io/amazon/index.html.

## Supplemental Information

Supplemental information for this article can be found online at http://dx.doi.org/10.7717/peerj-cs.2779#supplemental-information.

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
