# Peer review of "DSGRec: dual-path selection graph for multimodal recommendation"

_PeerJ Computer Science, doi:10.7717/peerj-cs.2779_

## Round 0.1 · original submission · Major Revisions

After carefully evaluating all the critical and constructive comments of the three Reviewers, I would ask the authors to go through them and fix their manuscript accordingly.

Reviewer 1 ·

Basic reporting

The paper presents the Dual-path Selective Graph Recommender (DSGRec), a novel approach for multimodal recommendation systems that addresses key limitations of existing methods. By decomposing user interaction signals into two distinct pathways—preference signals influenced by user interests and collaborative signals driven by similar user behaviors—DSGRec enhances the modeling of complex user-item interactions.

Experimental design

The experimental section of the paper evaluates the performance of the Dual-path Selective Graph Recommender (DSGRec) using three benchmark datasets. The authors conduct extensive experiments to compare DSGRec against several state-of-the-art recommendation models. The results demonstrate that DSGRec significantly outperforms these baselines in terms of recommendation accuracy and user satisfaction, validating its innovative dual-path approach and the effectiveness of its various components in capturing complex user-item interactions.

Validity of the findings

No Comment

Additional comments

The paper outlines an innovative approach with DSGRec; however, it would greatly benefit from providing specific examples of the hypergraph construction process and how hyperedges are utilized in signal enhancement would enhance comprehension.

Furthermore, a more in-depth discussion of the optimization techniques and hyperparameters used during training would help readers understand the model's tuning process. Including visual aids, such as flowcharts or diagrams, to illustrate the interaction between the various components

Explicitly stating the values chosen for key hyperparameters, such as learning rates, dropout rates, and the number of layers in the graph convolutional network, would provide clarity on the model's configuration.

Cite this review as

·

Basic reporting

The paper titled "DSGRec: Dual-path selection graph for multimodal recommendation" presents a novel approach to improve multimodal recommendation systems by addressing two significant limitations of existing methods. The authors introduce a Dual-path Selective Graph Recommender (DSGRec) that decomposes user interaction signals into two distinct pathways—preference signals and collaborative signals. This dual-path architecture aims to enhance the modeling of user behavior and improve the accuracy of recommendations by allowing for fine-grained collaboration between user behavior and multimodal information. Also describes the methodology employed, including the Heterogeneous Networks Information Collaboration (HNIC) module, which captures the dynamics of user-item interactions through a graph-based approach. Extensive experiments on benchmark datasets demonstrate the superiority of DSGRec over state-of-the-art recommendation baselines, establishing its effectiveness in delivering personalized recommendations. Some improvements are shown below:

1. The last paragraph of the introduction section, the fifth section, appears twice. Please confirm whether the second section 5 should be changed to section 7.
2. At the end of the experimental section of the paper, the adjustment of hyperparameters was mentioned, but the specific impact of hyperparameter selection on model performance was not clearly explained. Further exploration can be conducted on the influence of different hyperparameter settings on model performance.
3. The symmetric adjacency matrix A in the methodology section can be regarded as a formula and the format can be modified to be the same as the formula.
4. The references can be updated to include recently published related works
5. The conclusion part should be more refined to make the findings and contributions of the paper clearer. Further more, please note the difference between conclusion and abstract.
6. Add numbering to the references and indicate the corresponding numbering in the original citation table, which is clearer than using parentheses for citation.
7. It is suggested to avoid too many references at one place. For example “CF assumes users with similar interactions 40 share similar demands (He et al., 2017; Wang et al., 2019a; Xia et al., 2022a)”, please thoroughly review the manuscript and eliminate all clustered literature. This can be explained by mentioning 1 or 2 phrases in each citation to highlight its differences from others and why it is worth mentioning.
8. Images and tables must appear immediately after their first mention in the text. For example, the position in Figure 3 needs to be adjusted. Alternatively, before the first appearance of Figure 3, add corresponding prompts and reference explanations.

Experimental design

no comment

Validity of the findings

no comment

Cite this review as

Reviewer 3 ·

Basic reporting

This paper presents DSGRec, an approach to multi-modal recommendation systems by introducing a dual-path selection architecture, decomposing interaction information into collaborative and preference signals.
Pros:
1. The figures and tables of the paper effectively illustrate the concepts and results.
2. The methodology is explained in detail with appropriate mathematical notation.
3. The manuscript is well-structured and follows standard academic paper organization.
4. The writing is generally clear and easy to follow.
Cons:
1. The paper may not thoroughly discuss the computational complexity of the DSGRec model, especially when applied to large-scale datasets.
2. It is suggested to enhance linkage to recent literature that demonstrates the great potential and usefulness of different machine learning models
3. The paper could benefit from a more detailed discussion on the assumptions underlying the DSGRec model and its potential limitations. This includes how certain design choices might affect the model's performance and applicability.

Experimental design

Pros:
1. The research question is well-defined, relevant, and meaningful. The study addresses an identified knowledge gap by proposing a dual-path selection architecture for multi-modal recommendations.

Cons:
1. The paper should provide a comprehensive comparison with the latest state-of-the-art methods in multimodal recommendation.
2. The paper could include more sensitivity analysis to understand how changes in hyperparameters or model configurations impact the performance of DSGRec.
3. While the paper discusses the technical aspects of the model, it may not offer insights into the interpretability of the recommendations made by DSGRec. More case studies and visualization are needed.

Validity of the findings

1. The paper provides all underlying data, which are robust and statistically sound. The data are controlled and support the conclusions drawn by the authors.
2. The conclusions are well-stated and limited to the results, linking back to the original research question.

Additional comments

Please see the above comments.

Cite this review as

---

## Round 0.2 · accepted · Accept

I have carefully reviewed the revisions and I am pleased to confirm that you have adequately addressed all of the reviewers' comments.

I have personally evaluated the changes made and, after a thorough review, I am satisfied with the current version and believe that the revisions have substantially improved the manuscript, and I would consider it ready for publication.